# Factors Associated with the 30-Day and 1-Year Smoking Abstinence of Women in Korea: The Effect of Nicotine Dependency, Self-Efficacy, and Mental Illness

**DOI:** 10.3390/ijerph182111171

**Published:** 2021-10-24

**Authors:** Bo-Yoon Jeong, Min-Kyung Lim, Sang-Hwa Shin, Yu-Ri Han, Jin-Kyoung Oh, Hun-Jae Lee

**Affiliations:** 1National Cancer Center, Division of Cancer Prevention, National Cancer Control Institute, Goyang 10408, Korea; byjeong@ncc.re.kr (B.-Y.J.); quit@ncc.re.kr (S.-H.S.); jkoh@ncc.re.kr (J.-K.O.); 2Department of Social and Preventive Medicine, College of Medicine, Inha University, Incheon 22212, Korea; hyr333@hnamail.net; 3National Cancer Center, Department of Cancer Control and Population Health, Graduate School of Cancer Science & Policy, Goyang 10408, Korea

**Keywords:** woman smokers, sociodemographic features, health behaviors, smoking-related behaviors, smoking cessation

## Abstract

Despite the success of tobacco control efforts in reducing smoking rates during the past 50 years, data on the factors contributing to quitting success are still lacking. Smoking-related mortality among women has also not declined. Therefore, this study aimed to characterize sociodemographic features, smoking-related behaviors, mental illness, and smoking cessation in woman smokers in Korea who were registered in the Quitline program. Furthermore, factors associated with 30-day and 1-year successful smoking cessation after enrollment in the Quitline program were investigated. A total of 3360 adult woman Quitline users from 2007 to 2018 were included in the final analysis, with information on their age, education level, marital status, body mass index, frequency of alcohol consumption per month, nicotine dependency, self-efficacy for smoking cessation, presence of a smoking cessation supporter, and mental health history collected upon registration with the Quitline program in Korea. Their cessation outcome was investigated with a 1-year follow-up until the end of 2019. A multiple logistic regression analysis was conducted to identify factors associated with smoking cessation at the 30-day and 1-year follow-ups. The results of the multiple logistic regression analysis indicated that lower nicotine dependency, higher self-efficacy, and the presence of a smoking cessation supporter improved the odds of smoking cessation at the 30-day and 1-year follow-ups. In contrast, users with a mental health condition were less likely to achieve smoking cessation. Incorporating methods to increase self-efficacy in women who decide to quit smoking will contribute to facilitating more attempts to quit and achieving greater success in smoking cessation among woman smokers.

## 1. Introduction

Despite the success of tobacco control measures in several high-income countries in reducing the prevalence of smoking among women, globally approximately 1.8 million adult women are daily smokers. Moreover, over the past 50 years, the risk of mortality associated with women smoking has tripled and is now similar to that of men [1,2]. Furthermore, given the global epidemic of tobacco use and potential increases in marketing by tobacco companies, tobacco use among women may grow in Asian and African countries, including China, Thailand, Vietnam, Iran, Uganda, Cameroon, and Nigeria, where smoking among women is currently uncommon. Therefore, appropriate control measures should be adopted in Asian and African countries to prevent tobacco use and facilitate smoking cessation among women [1].

While in many Asian and African countries smoking among women is considered socially and culturally unacceptable, the prevalence of tobacco use among women is increasing. Moreover, there are a lack of studies in these regions focused on identifying factors associated with tobacco use among women or smoking cessation. Several studies conducted in Western countries have suggested that socioeconomic status, tobacco-use behaviors, nicotine sensitivity, and nicotine requirements differ between woman and man smokers [3,4,5,6,7]. Woman smokers may have a lower dependence on nicotine and thus use fewer cigarettes per day, prefer cigarettes with a lower nicotine content, and be less likely to deeply inhale cigarette smoke [8,9,10].

However, women seem to be less successful than men in smoking cessation. In part, this can be attributed to women’s greater propensity for smoking in order to alleviate negative emotions such as stress, anxiety, depression, anger, and loneliness, which increases their risk of relapse [3,10,11]. Although these psychosocial factors are probably much more important, the menstrual cycle influences cravings for cigarettes and the effect of pharmacotherapy drugs on smoking cessation among women, as hormonal decreases in estrogen and progesterone caused by cycle changes could possibly worsen withdrawal syndrome and increase activity in the neural circuits associated with craving [12]. In addition, nicotine replacement therapy (NRT) may be less effective in woman smokers owing to concerns about smoking cessation-associated weight gain, which can elevate their risk of relapse. Similar to studies conducted in Western populations, a few longitudinal population-based studies conducted in Asian and African countries have helped us to clarify the problem of smoking among women.

Since Korea’s enactment of the National Health Promotion Act in 1995, numerous anti-smoking policies and smoking cessation activities have been implemented. These include the designation of smoke-free areas, tobacco tax increases, and the enforcement of anti-tobacco campaigns. Moreover, smoking cessation services have been expanded, including the implementation of Quitline, smoking cessation clinics in 253 public health centers, and hospital-based smoking cessation clinics. Therefore, smoking rates among men have steadily declined (66.6% in 1998; 35.7% in 2019). In contrast, smoking rates among women have not substantially changed (6.5% in 1998; 6.7% in 2019), and woman lifetime smokers have begun smoking at younger ages (31.6 years in 1998; 24.1 years in 2015) [13]. Given the negative social norms associated with women smoking, woman smokers are less likely to disclose their smoking status, leading to the possibility that woman smoking rates have been under-reported [14]. Moreover, smoking stigma makes it difficult to reach woman smokers and offer appropriate cessation services, although women may experience more severe smoking cravings and diverse and stronger motivations for smoking compared to men; thus, support services for women are particularly critical [15].

Currently, there is a need for studies that identify factors associated with smoking and smoking cessation in woman smokers, which will aid in the development of effective smoking cessation interventions [16,17]. Therefore, this longitudinal study evaluated Quitline users in Korea for 1 year to identify factors that influenced the success of smoking cessation among woman smokers.

## 2. Methods

### 2.1. Quitline in Korea

A nationwide toll-free program called Quitline was launched in Korea in 2006 under the direction of the Ministry for Health, Welfare, and Family Affairs. Quitline is operated by the National Cancer Center and aims to decrease the prevalence of smoking and its related health burdens. At the user’s expense (the cost of a local phone call), smokers who wish to quit smoking can voluntarily contact Quitline by telephone, where they can register for the program. Once enrolled, smokers are offered systematic and comprehensive telephone-based behavioral counseling over a 1-year period free of charge. This includes 7 calls during the first 30 days of cessation. Those who achieve 30 days of cessation receive 14 additional calls over the next 11 months to support their continued cessation. Although NRT options are presented by Quitline coaches during the first counseling session, Quitline itself does not supply any kind of pharmacotherapy. However, Quitline does distribute booklets and send SMS messages to users in order to guide them through the cessation process. The communication scripts used by Quitline coaches were prepared by experts in tobacco control and health education. Intensive script and telecommunication training are mandatory for coaches when they are recruited, and their performance is monitored by a coordinator and an expert service quality committee. This committee decides on the contents and direction of additional training, if necessary [18,19].

### 2.2. Study Participants

Between 2007 and 2018, 7245 quit attempts were completed through Quitline in Korea by woman adult smokers, and they were followed up for 1 year until 31 December 2019. Among them, data on the first quit attempt of 4845 woman adult users were used for the study after excluding multiple enrollments due to repeated quit attempts during the follow-up periods. Subjects who did not try to quit after registering for Quitline (*n* = 907), were under 19 years of age (*n* = 435), or for whom we had incomplete information on the starting date of cessation (*n* = 2) were also excluded. A total of 3501 woman smokers were included in the analysis. Of these, 141 and 281 woman smokers were lost to follow-up at 30 days and 1 year, respectively. Therefore, 3360 and 3079 woman smokers were included in the final analysis in order to identify the factors associated with successful smoking cessation at 30 days and 1 year, respectively. This study was exempted from the review of the Institutional Review Board of the National Cancer Center of Korea.

### 2.3. Measures

Upon registration with Quitline, quitting coaches collected data on age (20–29, 30–39, 40–49, ≥50 years), education level (middle school or less, high school, college or more), marital status (single, married), body mass index (<18.5, 18.5–22.9, ≥23 kg/m^2^) [20], frequency of drinking alcohol per month (non-drinker, < 5 times, > 6 times), nicotine dependency score based on the Fagerström test (<3 (mild), 4–6 (moderate), 7–10 (severe)) [21], self-efficacy score for smoking cessation (0–2, 3–6, and 7–8) [22], presence of a smoking cessation supporter (no, yes), and having any of one or more mental illnesses (including depression, anxiety, sleep disorder, other psychologic diseases, manic-depressive disorder, panic disorder, etc.) (no, yes). Self-reported abstinence from smoking was measured at each follow-up counselling by coaches.

### 2.4. Statistical Analysis

The frequency distribution of the baseline characteristics of woman smokers who were included in the final analysis was found and the difference in characteristics between the subjects followed up and lost to follow-up was identified with the Mantel–Haenszel chi-square test.

The adjusted odds ratios (ORs) and 95% confidence intervals (CIs) were calculated for potential factors associated with the successful initiation and maintenance of smoking cessation from multiple logistic regression analyses adjusted for age only and adjusted for education level, marital status, Body Mass Index(BMI), frequency of drinking alcohol per month, nicotine dependency, self-efficacy for smoking cessation, presence of a smoking cessation supporter, and presence of a mental illness. Increasing and decreasing trends in the ORs across the strata of ordinal variables were identified by the Cochran–Armitage trend test. All statistical tests were conducted using SAS (SAS Institute, Inc., Cary, NC, USA), version 9.3.

## 3. Results

Woman smokers in their twenties and thirties with a high school education or greater, who drank alcohol more than six times per month, and who had smoking cessation supporters were prevalent within the study population. Only 23.1% and 16.13% of subjects had the lowest levels of self-efficacy for smoking cessation and the highest levels of nicotine dependency, respectively. Mental illness was present in 5.9% of all the subjects. Woman smokers who were lost to follow-up at 30-days and 1-year had a higher risk of unsuccessful smoking cessation, such as having a higher nicotine dependency, lower self-efficacy, or more serious mental illness (Table 1).

Only 24.4% and 8.3% of the subjects successfully completed the program and maintained smoking abstinence for 30 days and 1 year, respectively (Table 2 and Table 3). An elevated odds for achieving 30 days of successful smoking cessation was present in subjects with a lower nicotine dependency (7–10 vs. 4–6 OR: 1.18, 95% CI: 0.86–1.61; 7–10 vs. 0–3—OR: 1.70, 95% CI: 1.24–2.32), higher self-efficacy for smoking cessation (0–2 vs. 3–6—OR: 2.08, 95% CI: 1.54–2.80; 0–2 vs. 7–8—OR: 4.72, 95% CI: 3.36–6.61), and those with smoking cessation supporters (OR: 1.64, 95% CI: 1.31–2.05). Moreover, achieving 30 days of successful smoking cessation was associated with an increased level of education and frequency of alcohol drinking per month (Table 2).

Subjects with a college level of education or greater (less than middle school vs. college or more—OR: 2.27, 95% CI: 1.04–4.96) and greater self-efficacy for smoking cessation scores (0–2 vs. 3–6—OR: 1.93, 95% CI: 1.15–3.24; 0–2 vs. 7–8—OR: 3.59, 95% CI: 2.05–6.27) had significantly greater odds of successful smoking cessation at the 1-year follow-up. Moreover, increased education, having a lower nicotine dependency, and drinking less frequently were significant trends associated with smoking cessation at the 1-year follow-up (Table 3). By contrast, having a mental illness significantly decreased the odds of successful smoking cessation at 30 days (OR: 0.39, 95% CI: 0.22–0.72) and 1 year (OR: 0.22, 95% CI: 0.05–0.91) after the adjustment of all variables as appropriate (Table 2 and Table 3).

## 4. Discussion

The current study evaluated the short- and long-term abstinence of smoking among participants in their first quitting attempt who were enrolled in the telephone-based cessation program, Quitline, in Korea. It further identified factors that were associated with successful smoking cessation, such as lower nicotine dependency, higher self-efficacy, more education, higher frequency of drinking alcohol, and no presence of mental illness.

In Korea, the smoking prevalence among women is quite low (6.7% in 2019) and it is not socially acceptable [23]. Compared with the proportion of woman smokers among the total smokers (12.4%) reported in the National Health and Nutrition Examination Survey in 2015, the proportion of woman smokers who accessed and registered for Quitline (14.0%) was much higher [13,24]. This difference might be explained by the underreporting of woman smoking due to the social stigma around women smoking as well as the anonymity and confidentiality of Quitline services, which may allow woman smokers to use Quitline while avoiding stigma.

The successful quitting rate of woman smokers was identified as 24.4% and 8.3% for 30 days and 1 year, respectively, in the current study. This is comparatively lower than that in men (57.7% and 25.2% for 30-days and 1-year, respectively) based on the recent data reported from Korean Quitline [24]. This reflects similar results reported by previous studies which indicated that negative emotions (i.e., a result of social stigma) increased the risk of relapse among woman smokers [3,10,11].

In the present study, higher self-efficacy and lower nicotine dependency most strongly increased the odds of successful cessation at 30 days among woman smokers, which is consistent with previous studies. Furthermore, we confirmed that higher self-efficacy for smoking cessation and lower nicotine dependency were associated with the maintenance of long-term smoking cessation. Self-efficacy is an important psychological determinant of smoking cessation. However, woman smokers are less confident in their ability to successfully quit smoking and have more difficulty in coping with smoking temptations [25]. Even if woman smokers have a lower nicotine dependency than their male counterparts, their self-efficacy for quitting is lower than that of male smokers [26]. It is well known that greater nicotine dependencies can exacerbate withdrawal symptoms, particularly during the initial stage of cessation, and could thus contribute to a relapse [27]. Therefore, greater nicotine dependencies may contribute to failure in smoking cessation both in the initial stages and in the long term. More importantly, increasing self-efficacy for cessation should be prioritized in cessation interventions that target woman smokers.

Woman smokers with greater levels of education were more likely to succeed in smoking cessation, and this effect was stronger with long-term cessation. This result is consistent with that of previous studies, where smokers with greater levels of education were found to be more likely to attempt to quit smoking [28,29]. In general, smokers with a higher level of education may be better equipped to access appropriate smoking cessation information. Moreover, based on having higher self-efficacy and a greater capacity to make decisions regarding the health benefits of smoking cessation, they may be more confident win undertaking positive behavioral changes [19]. This study identified that woman smokers who had a smoking cessation supporter were more successful in their initial cessation. Positive social support is an important factor that contributes toward successful smoking cessation in both man and woman smokers but is more important in women. Previous works have indicated that positive social relationships and understanding and support from others were much more important for woman smokers attempting to quit smoking [30,31,32]. In contrast, self-mastery and insight were more important for man smokers attempting to quit [30,31,32].

In addition, this study determined that woman smokers who drank alcohol over six times per month were prevalent. More frequent drinking was associated with reduced odds of successful smoking cessation. As suggested by previous studies, drinking alcohol is an addictive behavior, and, when combined with smoking, could increase one’s dependency on smoking. Frequent alcohol drinking could also be considered as a predictor of negative social relationships [33] or as a negative behavior that releases individuals from the social pressures or negative emotions associated with smoking. Therefore, smoking cessation interventions could consider incorporating non-smokers who are socially close to the smoker to provide emotional support in order to aid smokers in their attempt to quit smoking. The incorporation of social support is particularly important for woman smokers.

Smoking in women is known to be closely related to high-arousal negative emotions, such as stress, anxiety, and anger, as well as low-arousal negative emotions, such as sadness, depression, lack of energy, and loneliness [11,34]. Therefore, it is not surprising that woman smokers with negative emotions are less likely to be successful in smoking cessation, which is consistent with our data. In total, 5.6% of woman smokers in our study had a mental illness. Of these, only 12.6% achieved smoking cessation by 30 days, which decreased to 3.7% at the 1-year follow-up. Successful smoking cessation was 60% lower in woman smokers with a mental illness than those without. This indicates that more intense and careful approaches should be considered for woman smokers with mental illness in order to improve their success in smoking cessation.

While enrolling woman smoking subjects from Quitline is effective for reaching woman smokers, this study suffers from several limitations. First, the subjects were not sampled from the general woman smoking population and instead were recruited from those who voluntarily registered to Quitline. Moreover, some of them were lost to follow-up. There is therefore potential for selection bias. However, woman smokers lost to follow-up did not distort the study results, as we could obtain similar results from the analysis by including them as unsuccessful cessation cases (see Appendix A). Second, as the research data were obtained using the information collected in the initial registration, other factors related to the success of smoking cessation relating to the effectiveness of the Quitline counseling programs and the number of previous smoking cessation attempts were not considered. Finally, it is possible that the estimated smoking cessation success rate represents an overestimate because self-reported cessation outcomes were not validated.

Using a longitudinal follow-up design, this study identified factors associated with successful smoking cessation among woman smokers in Korea. In particular, interventions in woman smokers should focus on increasing self-efficacy for smoking cessation, even among smokers with low nicotine dependency, and ensure positive social support, which can enable the initiation and maintenance of smoking cessation. This study further identified characteristics associated with a higher risk of relapse among woman smokers. These include lower levels of education, drinking more frequently, and having a mental illness. The results are informative for many Asian countries, where there is social stigma around woman smoking and the prevalence of women who smoke is low. Our study could therefore help to improve the efficacy of smoking cessation, specifically in woman smokers, which can often differ from that of similarly situated man smokers.

## Figures and Tables

**Table 1 ijerph-18-11171-t001:** Baseline characteristics of all woman smokers, at 30-days and 1-year follow-up, and those lost to follow-up.

Variables	Total	Followed Up for 30-Day after Starting Quit	Follow-Up Loss	*p*-Value ^a^	Followed up for 1-Year after Starting Quit	Follow-Up Loss	*p*-Value ^b^
	*n* = 3501	*n* = 3360	*n* = 141		*n* = 3079	*n* = 422	
Age groups (years)							
20–29	1152 (32.9)	1127 (33.5)	25 (17.7)	<0.0001	1035 (33.6)	117 (27.7)	<0.0001
30–39	1223 (34.9)	1185 (35.3)	38(27.0)		1112 (36.1)	111 (26.3)	
40–49	657 (18.8)	619 (18.4)	38 (27.0)		557 (18.1)	100 (23.7)	
≥50	469 (13.4)	429 (12.8)	40 (28.4)		375 (12.2)	94 (22.3)	
Education level				0.1894			0.0067
Middle school or less	319 (9.3)	302 (9.1)	17 (12.8)		276 (9.1)	43 (10.6)	
High school	1746 (50.8)	1675 (50.7)	71 (53.4)		1516 (50.0)	230 (56.7)	
College or more	1375 (40.0)	1330 (40.2)	45 (33.8)		1242 (40.9)	133 (32.8)	
Marital status				0.2232			0.7812
Single	1726 (49.3)	1664 (49.5)	62 (44.3)		1521 (49.4)	205 (48.7)	
Married	1773 (50.7)	1695 (50.5)	78 (55.7)		1557 (50.6)	216 (51.3)	
Body mass index (kg/m^2^)				0.0321			0.0001
Underweight (<18.5)	557 (16.0)	534 (16.0)	23 (16.5)		495 (16.1)	62 (14.8)	
Normal (18.5–22.9)	1994 (57.2)	1928 (57.6)	66 (47.5)		1786 (58.2)	208 (49.8)	
Overweight or more (≥23.0)	937 (26.9)	887 (24.5)	50 (36.0)		789 (25.7)	148 (35.4)	
Frequency of drinking alcohol per month				0.4715			<0.0001
Non-drinker	571 (20.6)	557 (20.7)	14 (16.3)		554 (22.0)	17 (6.6)	
Less than 5 times	1014 (36.6)	978 (36.4)	36 (41.9)		911 (36.2)	103 (39.8)	
Over 6 times	1189 (42.9)	1153 (42.9)	36 (41.9)		1050 (41.8)	139 (53.7)	
Nicotine dependency				<0.0001			<0.0001
0–3 (mild)	1303 (37.9)	1276 (38.7)	27 (19.3)		1193 (39.4)	110 (27.2)	
4–6 (moderate)	1582 (46.0)	1508 (45.7)	74 (52.9)		1376 (45.4)	206 (50.9)	
7–10 (severe)	552 (16.1)	513 (15.6)	39 (27.9)		463 (15.3)	89 (22.0)	
Self-efficacy for smoking cessation				0.0071			0.0569
0–2	784 (23.1)	738 (22.6)	46 (34.1)		672 (22.4)	112 (27.7)	
3–6	2027 (59.6)	1956 (59.9)	71 (52.6)		1799 60.0)	228 (56.4)	
7–8	591 (17.4)	573 (17.5)	18 (13.3)		527 (17.6)	64 (15.8)	
Presence of quit supporter				0.4625			0.0582
No	986 (28.6)	946 (28.5)	40 (31.5)		864 (28.1)	122 (32.8)	
Yes	2461 (71.4)	2374 (71.5)	87 (68.5)		2211 (71.9)	250 (67.2)	
Presence of a mental illness				0.0001			<0.0001
No	3192 (94.1)	3087 (94.4)	105 (86.1)		2867 (94.7)	325 (89.3)	
Yes	200 (5.9)	183 (5.6)	17 (13.9)		161 (5.3)	39 (10.7)	

^a^ Mantel–Haenszel chi-square test for the difference in frequency distribution between subjects followed up for 30 days and lost to follow-up. ^b^ Mantel–Haenszel chi-square test for the difference in frequency distributions between subjects followed up for 1 year and lost to follow-up.

**Table 2 ijerph-18-11171-t002:** Odds ratios and 95% confidence intervals of potential factors associated with smoking cessation at the 30-day follow-up.

Variables	Total [N]	Successful Cessation ^c^ [N (%)]	OR (95% CI) ^d^	OR (95% CI) ^e^
	3360	818 (24.4)		
Age groups (years)				
≥50	429	110 (25.6)	Ref	Ref
40–49	619	118 (19.1)	0.68 (0.51–0.92)	0.75 (0.50–1.13)
30–39	1185	307 (25.9)	1.01 (0.79–1.31)	1.01 (0.69–1.48)
20–29	1127	283 (25.1)	0.97 (0.75–1.26)	0.89 (0.59–1.33)
P for trend ^f^			P: 0.2292	P: 0.9519
Education level				
Middle school or less	302	66 (21.9)	Ref	Ref
High school	1675	372 (22.2)	1.08 (0.78–1.48)	1.19 (0.79–1.81)
College or more	1330	367 (27.6)	1.44 (1.03–2.00)	1.45 (0.94–2.24)
P for trend ^f^			P: 0.0021	P: 0.0152
Marital status				
Married	1695	394 (23.2)	Ref	Ref
Single	1664	424 (25.5)	1.09 (0.90–1.31)	1.22 (0.96–1.54)
Body mass index (kg/m^2^) ^a^				
Normal (18.5–22.9)	1928	463 (24.0)	Ref	Ref
Underweight (<18.5)	534	120 (22.5)	0.88 (0.70–1.11)	0.83 (0.63–1.10)
Overweight or more (≥23.0)	887	231 (26.0)	1.14 (0.95–1.37)	1.25 (0.99–1.58)
P for trend ^f^			P: 0.0553	P: 0.0108
Frequency of drinking alcohol per month				
Non-drinker	557	141 (25.3)	Ref	Ref
Less than 5 times	978	270 (27.6)	1.11 (0.87–1.41)	0.97 (0.75–1.26)
Over 6 times	1153	250 (21.7)	0.80 (0.63–1.02)	0.78 (0.60–1.01)
P for trend ^f^			P: 0.0165	P: 0.0243
Nicotine dependency ^b^				
7–10 (severe)	513	83 (16.2)	Ref	Ref
4–6 (moderate)	1508	319 (21.2)	1.39 (1.06–1.81)	1.18 (0.86–1.61)
0–3 (mild)	1276	390 (30.6)	2.26 (1.73–2.94)	1.70 (1.24–2.32)
P for trend ^f^			P: <0.0001	P: <0.0001
Self-efficacy for smoking cessation				
0–2	738	91(12.3)	Ref	Ref
3–6	1956	467(23.9)	2.21 (1.73–2.82)	2.08 (1.54–2.80)
7–8	573	237(41.4)	5.01 (3.80–6.60)	4.72 (3.36–6.61)
P for trend ^f^			P: <0.0001	P: <0.0001
Presence of a smoking cessation supporter				
No	946	175 (18.5)	Ref	Ref
Yes	2374	629 (26.5)	1.59 (1.32–1.92)	1.64 (1.31–2.05)
Presence of a mental illness				
No	3087	768 (24.9)	Ref	Ref
Yes	183	23 (12.6)	0.45 (0.29–0.70)	0.39 (0.22–0.72)

Note. OR = odds ratios; CI = confidence interval; P = P for trend. ^a^ Classification of BMI using the guideline for the Asian population. ^b^ Fagerstrom test for nicotine dependence score. ^c^ Includes participants who maintained cessation for more than 30 days and those who maintained cessation but had not yet completed the 30-day program period. ^d^ Multiple logistics regression models adjusted for age. ^e^ Multiple logistics regression models adjusted for age, education level, marital status, BMI, frequency of drinking alcohol per month, nicotine dependency, self-efficacy for smoking cessation, presence of a smoking cessation supporter, and presence of a mental illness. ^f^ Cochran–Armitage trend test.

**Table 3 ijerph-18-11171-t003:** Odds ratio and 95% confidence intervals of potential factors associated with smoking cessation at the 1-year follow-up.

Variables	Total [N]	Successful Cessation ^c^ [N %]	OR (95% CI) ^d^	OR (95% CI) ^e^
	3079	255 (8.3%)		
Age groups (years)				
≥50	375	30 (8.0)	Ref	Ref
40–49	557	32 (5.8)	0.70 (0.42–1.18)	0.66 (0.33–1.32)
30–39	1112	96 (8.6)	1.09 (0.71–1.67)	1.03 (0.56–1.93)
20–29	1035	97 (9.4)	1.19 (0.78–1.82)	1.16 (0.60–2.23)
P for trend ^f^			P: 0.0718	P: 0.2002
Education level				
Less than middle school	276	17 (6.2)	Ref	Ref
High school	1516	108 (7.1)	1.18 (0.67–2.07)	1.59 (0.74–3.41)
College or more	1242	126 (10.1)	1.75 (0.98–3.12)	2.27 (1.04–4.96)
P for trend ^f^			P: 0.0070	P: 0.0077
Marital status				
Married	1557	121 (7.8)	Ref	Ref
Single	1521	134 (8.8)	0.99 (0.73–1.35)	0.96 (0.66–1.39)
Body mass index (kg/m^2^) ^a^				
Normal (18.5–22.9)	1786	146 (8.2)	Ref	Ref
Underweight (<18.5)	495	40 (8.1)	0.93 (0.64–1.35)	0.92 (0.60–1.42)
Overweight or more (≥23.0)	789	68 (8.6)	1.11 (0.82–1.50)	1.22 (0.84–1.76)
P for trend ^f^			P: 0.4053	P: 0.2799
Frequency of drinking alcohol per month				
Non-drinker	554	46 (8.3)	Ref	Ref
Less than 5 times	911	90 (9.9)	1.16 (0.80–1.69)	1.08 (0.73–1.59)
Over 6 times	1050	65 (6.2)	0.69 (0.46–1.03)	0.68 (0.45–1.04)
P for trend ^f^			P: 0.0227	P: 0.0270
Nicotine dependency ^b^				
7–10 (severe)	463	21 (4.5)	Ref	Ref
4–6 (moderate)	1376	92 (6.7)	1.50 (0.92–2.43)	1.04 (0.62–1.75)
0–3 (mild)	1193	137 (11.5)	2.65 (1.65–4.27)	1.47 (0.88–2.46)
P for trend ^f^			P: <0.0001	P: 0.0275
Self-efficacy for smoking cessation				
0–2	672	26 (3.9)	Ref	Ref
3–6	1799	143 (8.0)	2.11 (1.37–3.23)	1.93 (1.15–3.24)
7–8	527	78 (14.8)	4.25 (2.68–6.74)	3.59 (2.05–6.27)
P for trend ^f^			P: <0.0001	P: <0.0001
Presence of a smoking cessation supporter				
No	864	55 (6.4)	Ref	Ref
Yes	2211	199 (9.0)	1.45 (1.06–1.98)	1.23 (0.87–1.75)
Presence of a mental illness				
No	2867	248 (8.7)	Ref	Ref
Yes	161	6 (3.7)	0.44 (0.19–1.00)	0.22 (0.05–0.91)

Note. OR = odds ratios; CI = confidence interval; P = P for trend. ^a^ Classification of BMI using guidelines for the Asian population 20-Pacific. ^b^ Fagerstrom test for nicotine dependence score. ^c^ Includes participants who maintained cessation for more than 1 year and those who maintained cessation but had not yet completed the 1-year program period. ^d^ Multiple logistics regression models adjusted for age. ^e^ Multiple logistics regression models adjusted for age, education level, marital status, BMI, frequency of drinking alcohol per month, nicotine dependency, self-efficacy for smoking cessation, presence of a smoking cessation supporter, and presence of a mental illness. ^f^ Cochran–Armitage trend test.

## Data Availability

Restrictions apply to the availability of these data. Data was obtained from National Quitline and are available with the permission of Korean government.

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
