# Peer review of "Factors Associated with the 30-Day and 1-Year Smoking Abstinence of Women in Korea: The Effect of Nicotine Dependency, Self-Efficacy, and Mental Illness"

_ijerph, 2021, doi:10.3390/ijerph182111171_

Round 1

Reviewer 1 Report

The paper entitled “The Factors Associated With 30 Days and 1-Year Smoking 2 Abstinence of Women in Korea: The Effect of Nicotine 3 Dependency, Self-Efficacy, and Mental Illness” by Jeong et al. was an informative read.

By using data from a smoking cessation program, Quitline, in Korea, they have identified factors associated with short- and long-term factors associated with smoking cessation among women. Factors associated were quit level of nicotine dependence, self-efficacy for smoking cessation, smoking cessation support, and mental health history. Relevant associations were discussed in enough details. The authors showed sensitivity to the cultural and gender norms in Asia with regards to smoking for women. They have set up the premise and importance of the study well in the introduction.

Some pointers where I thought could improve the manuscript.

  1. Despite the potential bias in their data collection method, the authors have the advantage of longitudinal data of their respondents. Have they considered exploring factors predictive of successful smoking cessations in their sample? This piece of data could potentially set their paper apart from many cross-sectional survey papers.
  2. I am interested in what kind of mental illnesses were associated with poorer smoking cessations outcome in their sample. Was information about the type of mental illness captured in their survey? This could inform clinicians which clinical population group(s) to pay more attention when it comes to smoking cessation in practice.
  3. It may be helpful to include a summary (with more details of the factors associated) at the start of the discussion section to wrap up the description of the preceding results section and set up for your discussion as you delve into each associated factor.
  4. Descriptions of the methodology segment was sufficient, although more information on the statistical analysis will allow me to be more critical of the data analysis component of the paper.
  5. The total numbers for the variables for your data tables do not always tally with the given total of n=3360 or n=818 and n=3079 or n=255. Pls identify the number of missing data and explain in the methods what you do with them in the analysis.

Reviewer 2 Report

The manuscript assessed factors associated with 30 days and 1-year smoking abstinence among 3360 female individuals who smoke cigarettes in Korea. While the study design and results were straightforward and reasonable, I will discuss a few issues. 

The manuscript started with a broad discussion of cigarette smoking in Asia and Africa, which is fine, but the authors should also briefly describe epidemiology of cigarette smoking (prevalence, trend, cessation rates etc) in Korea, especially among females, to situate the manuscript within an appropriate context.  

Please describe how abstinence was assessed. Was abstinence confirmed by objective evaluation, such as urine cotinine test or expired-air CO level?

Women who lost to follow-up could be quite different from those who remained in the study. It is likely that they could have a high risk of unsuccessful smoking cessation. Thus, simply excluding respondents who lost to follow-up could introduce bias. At least, the authors could compare the baseline characteristics between these two groups.  

Please clarify the rationale of excluding adults aged 18 and 19 years?

The participants were enrolled during a 12-year period. Why was a time trend not considered in the analytic strategy?  

Please include a descriptive Table, presenting baseline characteristics of respondents overall, and stratifying by those who achieved abstinence at 30-day follow-up, and those who achieved abstinence at 1-year.

The authors mentioned that they excluded respondents who had incomplete information, why the numbers for covariates do not added up to the total analytic sample?! For example, the numbers of respondents with three education levels were 302, 1675, and 1330, respectively (Table 1). So, the total number should be 3307, does that mean 53 respondents had missing value on this variable? A few other variables also had similar issue.

A few minor comments: The independent variables in the Tables should be better aligned, and the columns reporting ORs should be better labeled for easier interpretation (e.g., age-adjusted OR, multivariable-adjusted OR).   

Round 2

Reviewer 2 Report

The authors have done a fine job addressing my comments. I only have one remaining question - please clarify the approach in handling missing data on covariates. It seems that age-adjusted OR and multivariable adjusted OR were calculated based on different analytic samples if respondents with missing data on covariates were dropped with list-wise deletion. The sample size for multivariable adjusted ORs should be smaller than indicated by total N in Tables 2 and 3. 

Author Response

Response to the comment from reviewer 2

Point 1 : The authors have done a fine job addressing my comments. I only have one remaining question - please clarify the approach in handling missing data on covariates. It seems that age-adjusted OR and multivariable adjusted OR were calculated based on different analytic samples if respondents with missing data on covariates were dropped with list-wise deletion. The sample size for multivariable adjusted ORs should be smaller than indicated by total N in Tables 2 and 3. 

Response 1 : Thanks for your question. As you guessed, we have small number of missing data in each variable except variable 'age'. Therefore, missing data of each variable would be excluded, when the multiple logistic regression analysis adjusted for age and all variables as appropriate  were applied. It means that the sample size for multivariable adjusted ORs should be smaller than indicated by total N in Tables 2 and 3, as you mentioned.